# Multi-Omics Profiling to Assess Signaling Changes upon VHL Restoration and Identify Putative VHL Substrates in Clear Cell Renal Cell Carcinoma Cell Lines

**DOI:** 10.3390/cells11030472

**Published:** 2022-01-29

**Authors:** Xuechun Wang, Jin Hu, Yihao Fang, Yanbin Fu, Bing Liu, Chao Zhang, Shan Feng, Xin Lu

**Affiliations:** 1Fundamental Research Center, Shanghai YangZhi Rehabilitation Hospital (Shanghai Sunshine Rehabilitation Center), School of Life Sciences and Technology, Tongji University, Shanghai 200092, China; xwang53@nd.edu (X.W.); ianbin_fu@163.com (Y.F.); 2Department of Biological Sciences, Boler-Parseghian Center for Rare and Neglected Diseases, Harper Cancer Research Institute, University of Notre Dame, Notre Dame, IN 46556, USA; 3Mass Spectrometry & Metabolomics Core Facility, Key Laboratory of Structural Biology of Zhejiang Province, Westlake University, Hangzhou 310024, China; hujing@westlake.edu.cn; 4Department of the Applied and Computational Mathematics and Statistics, University of Notre Dame, Notre Dame, IN 46556, USA; yfang5@nd.edu; 5Department of Urology, Eastern Hepatobiliary Surgery Hospital, Shanghai 201805, China; 13501616398@163.com

**Keywords:** von Hippel–Lindau, VHL, clear cell renal cell carcinoma, ubiquitome, E3 ubiquitin ligase, TGFBI, NFKB2, 786-O, multi-omics

## Abstract

The inactivation of von Hippel–Lindau (VHL) is critical for clear cell renal cell carcinoma (ccRCC) and VHL syndrome. VHL loss leads to the stabilization of hypoxia-inducible factor α (HIFα) and other substrate proteins, which, together, drive various tumor-promoting pathways. There is inadequate molecular characterization of VHL restoration in VHL-defective ccRCC cells. The identities of HIF-independent VHL substrates remain elusive. We reinstalled VHL expression in 786-O and performed transcriptome, proteome and ubiquitome profiling to assess the molecular impact. The transcriptome and proteome analysis revealed that VHL restoration caused the downregulation of hypoxia signaling, glycolysis, E2F targets, and mTORC1 signaling, and the upregulation of fatty acid metabolism. Proteome and ubiquitome co-analysis, together with the ccRCC CPTAC data, enlisted 57 proteins that were ubiquitinated and downregulated by VHL restoration and upregulated in human ccRCC. Among them, we confirmed the reduction of TGFBI (ubiquitinated at K676) and NFKB2 (ubiquitinated at K72 and K741) by VHL re-expression in 786-O. Immunoprecipitation assay showed the physical interaction between VHL and NFKB2. K72 of NFKB2 affected NFKB2 stability in a VHL-dependent manner. Taken together, our study generates a comprehensive molecular catalog of a VHL-restored 786-O model and provides a list of putative VHL-dependent ubiquitination substrates, including TGFBI and NFKB2, for future investigation.

## 1. Introduction

Kidney cancer, or renal cell carcinoma (RCC), affects nearly 300,000 individuals and causes over 130,000 deaths annually worldwide [1]. The incidence of RCC has been increasing over several decades with the higher prevalence of obesity as one potential contributing factor [1]. Clear cell renal cell carcinoma (ccRCC) is the most common (~75%) and lethal form of RCC [1]. ccRCC occurs in both sporadic and familial forms. Patients affected by von Hippel–Lindau (VHL) disease are at high risk of developing multifocal, bilateral ccRCC [2]. In sporadic ccRCC, frequent (~90%) mutations that inactivate VHL or the deletion of the tumor suppressor gene *VHL* is an obligatory initiating step in the carcinogenesis process [3]. 

VHL is the substrate recognition component of the Cul2-Rbx1-EloBC-VHL E3 ubiquitin ligase complex [4]. VHL binds hydroxylated target proteins in an oxygen-dependent manner, leading to their polyubiquitination and degradation. The most characterized VHL substrates are the transcription factors hypoxia-inducible factors 1α (HIF1α) and HIF2α [5]. Loss of VHL stabilizes HIF1α and HIF2α, which results in the transcriptional activation of HIF targets that promote angiogenesis (e.g., VEGFA, PDGF), metabolic reprograming toward the Warburg effect (e.g., GLUT1, hexokinase 2, LDHA), cell proliferation (e.g., TGFα, EGFR, NF-κB) and other malignancy-associated traits [5,6,7]. In the context of ccRCC, HIF2α plays a major tumor-promoting role whereas HIF1α appears to function as a tumor suppressor [8,9,10,11]. This prompts the development of HIF2α-specific inhibitor Belzutifan (MK-6482), which was recently approved to treat patients with VHL disease tumors [12,13,14,15]. It is already known that a significant fraction of ccRCC cases remain resistant to HIF2α-inhibitor treatment [12,13,14], highlighting the importance of identifying additional therapeutic vulnerabilities in ccRCC.

The resistance of *VHL*-defective ccRCC to the HIF2α inhibitor may be at least partly mediated through the activity of VHL to recognize and regulate the stability or activity of non-HIFα substrate proteins. A few non-HIFα substrates of VHL have been identified, including AKT [16], fibronectin [17], collagen IV [18], AURKA [19], RPB1 [20], ZHX2 [21] and SFMBT1 [22]. The dysregulation of these non-HIFα targets in the absence of VHL contributes to the tumorigenesis of ccRCC [3]. Clearly, there is a strong need to gain a deeper understanding of VHL function and find other VHL targets in order to develop new therapies for ccRCC and overcoming the resistance to the HIF2α inhibitor. To address this, we have profiled and integrated transcriptome, proteome and ubiquitome of a classical model of *VHL*-defective ccRCC, 786-O cell line (harboring *VHL* frameshift mutation at codon 104), after restoring VHL expression. The data support that the predominant effect of VHL restoration is the downregulation of HIF-related signaling and metabolic pathways. More importantly, through the co-analysis of our data and the Clinical Proteomic Tumor Analysis Consortium (CPTAC) study of ccRCC [23], 57 proteins were identified that are ubiquitinated and downregulated by VHL restoration in 786-O and overexpressed in ccRCC clinical samples. Higher expression of this gene signature was significantly associated with worse prognosis in ccRCC patients. 

Among the 57 proteins, we have focused on two VHL substrate candidates, transforming growth factor beta induced (TGFBI) and nuclear factor kappa B subunit 2 (NFKB2), both of which are likely important players in promoting ccRCC. TGFBI (BIGH3), a secreted extracellular matrix protein, regulates various biological functions, including cell adhesion and bone formation [24]. In ccRCC, TGFBI plays an important role in promoting bone metastasis through the suppression of osteoblast differentiation [25]. TGFBI protein overexpression is strongly associated with a more advanced tumor stage, metastasis and cancer-specific mortality in ccRCC patients [26]. It is known that VHL represses TGFBI expression in an HIF-independent manner and the loss of VHL results in a higher TGFBI level [27,28], suggesting that TGFBI is a direct substrate of VHL. NFKB2 is a key component of the non-canonical NF-κB pathway, and this pathway activity (assessed by NIK, IKKα and RelB staining) was shown to be associated with poor survival in ccRCC patients [29]. When the non-canonical NF-κB signaling is activated, the NFKB2 precursor protein p100 is phosphorylated and polyubiquitinated for the selective degradation and generation of p52, and the p52/RelB complex enters the nucleus as a transcriptional activator [30]. Processing of the NFKB2/p100 precursor to p52 is through polyubiquitination at lysine residue 855 (K855) by the SCF^β-TrCP^ E3 ubiquitin ligase [31]. VHL loss upregulates the canonical NF-κB pathway through HIF-dependent and independent mechanisms [21,32,33]; however, its role in the non-canonical NF-κB pathway is unknown.

Overall, through multi-omics studies, we aimed to generate a list of putative non-HIFα targets for VHL, uncover potentially additional functional mechanisms of VHL and illuminate new therapeutic opportunities with *VHL*-deficient ccRCC. 

## 2. Materials and Methods

### 2.1. Cell Culture

786-O and HEK293T were purchased from American Type Culture Collection (ATCC, CRL-1932, CRL-3216). VHL restoration was achieved by the retroviral expression of the HA-VHL/wt-pBabe-puro vector (Addgene, #19234) [34]. Triple-reporter (TR) labeling (GFP, luciferase, thymidine kinase) was achieved by the stable expression of the SFG-_NES_TGL retroviral vector [35]. These cell lines were cultured in RPMI1640 medium (GE Healthcare, SH30027.01) supplemented with 10% fetal bovine serum (GE Healthcare, SH30396.03) and 100 U/mL of Penicillin–Streptomycin (Caisson Labs, PSL01) at 37 °C in a humidified incubator with 5% CO_2_. All cells were tested with the Mycoplasma Assay Kit (Agilent Technologies, 302109) and tested as free of mycoplasma. 

### 2.2. Colony and Sphere Formation Assays

For the 2D colony formation assay, 500 cells per well were seeded into 6-cm plates and incubated for 7 days with the medium refreshed every 2–3 days. Cells were fixed and stained with 0.2% crystal violet in 25% methanol solution for 1 h at room temperature, and the total colony number per well was counted. For the 3D tumor sphere formation assay, 500 cells per well were seeded on Day 0 into Corning 96-well Clear Flat Bottom Ultra-Low Attachment Microplates. Plates were incubated for 7 days with the medium refreshed every 2–3 days. Spheroids were counted under a microscope. 

### 2.3. Orthotopic Tumor Model

All animal works performed in this study were approved by the Institutional Animal Care and Use Committee at the University of Notre Dame. NCr nude female or male mice (6–8 weeks old) were purchased from Taconic Biosciences (Rensselaer, NY, USA). For orthotopic modeling, 2 × 10^6^ viable cells were injected near the lower pole into the renal parenchyma. Tumors were monitored weekly by bioluminescent imaging with the Spectral Ami HT Advanced Molecular Imager (Spectral Instruments Imaging, Tucson, AZ, USA). Mice were sacrificed at the indicated endpoint. Tumors were preserved in 10% formalin followed by paraffin embedding and standard hematoxylin and eosin (H&E) staining. 

### 2.4. Western Blotting 

The Western blot procedure was conducted as we described [36]. Briefly, cells were lysed in RIPA buffer containing protease inhibitor and phosphatase inhibitor. For secreted TGFBI detection, 786-O-Ctrl and 786-O-VHL cells were seeded in 10 cm dishes. After reaching 80% confluence, cells were cultured in a serum-free medium for 12 h, followed by medium collection and concentration by 50-fold using Amicon Ultra-4 50K Centrifugal Filter Units (Millipore Sigma, UFC801008). All samples were run through standard SDS-PAGE. Primary antibodies included VHL (Cell Signaling Technology, #68547), HIF2α (Novus Biologicals, #NB100-122), VEGFA (Abcam, #ab46154), GLUT1 (Novus Biologicals, #NB110-39113SS), β-actin (Santa Cruz, #sc-47778), TGFBI (Cell Signaling Technology, #5601T) and NFKB2 (Cell Signaling Technology, #4882T). Secondary antibodies included HRP-linked anti-rabbit IgG (Cell Signaling Technology, #7074) and HRP-linked anti-mouse IgG (Cell Signaling Technology, #7076V). 

### 2.5. Co-Immunoprecipitation 

Cell lysates were immunoprecipitated overnight with 1:100 diluted NFkB2 antibody (Cell Signaling Technology, #4882T). Next, the lysate was incubated with Protein G Sepharose (GE-Healthcare, #GE17-0618) for 1 h, and washed three times with cell lysis buffer before being resuspended in loading dye and boiled for 5 min at 95 °C. Eluted proteins were immunoblotted as described above. 

### 2.6. Cell Adhesion Assay

For adhesion assays, 96-well plates were coated with fibronectin (Sigma-Aldrich, F4759) for 2 h and then air-dried. 2 × 10^4^ cells were dispensed into each well of the 96-well plate and then incubated for 1 h. Cells were gently washed three times with PBS to remove the unattached cells. Resazurin assays were used to quantify the number of attached cells in samples. Absorbance at 570 nm was measured on a spectrophotometer.

### 2.7. Targeted Mutagenesis of NFKB2

NFKB2 plasmid (Addgene, #174734) was used as template for lysine site mutation. K72 was mutated to arginine (K72R) by a standard site-directed mutagenesis PCR technique with primers cggtgcctccagtgagaggggccga and ggcagtcctccatgggaggggcc.

### 2.8. RNA Sequencing

RNA was extracted in duplicates from 786-O-Ctrl and 786-O-VHL cells using the Quick-RNA Miniprep Kit (Zymo Research, #R1054) following the manufacturer’s manual. RNA quality was assessed with Agilent Bioanalyzer 2100. Purified RNA (1 μg) was used for the Illumina NovaSeq 6000 Sequencing (Novagene). RNA-seq sequencing reads were aligned to hg19, and the DESeq2 package [37] was used to calculate differential expression genes. Pathway analysis was performed using Gene Set Enrichment Analysis (GSEA) [38]. Gene list enrichment was analyzed by MetaCore (version 21.3 build 70600) or Enrichr [39,40].

### 2.9. Proteomics

The cell pallet was collected and lysed by RIPA buffer to exact the proteins. The protein concentrations were determined by the BCA method and 100 μg of proteins from each sample were acquired for the further digestion. After reduction and alkylation, the proteins were precipitated by cold acetone, followed by resolving with 100 mM of TEAB buffer and trypsin digestion at 37 °C overnight. Then the tryptic peptides were labeled with TMT reagent and the reaction was terminated by hydroxylamine. After mixing the TMT-labeled peptides from different groups (e.g., control and *VHL* overexpression), the mixture was desalted by HLB column and then separated into 10 fractions using the high-pH reverse phase fractionation approach.

For liquid chromatography with tandem mass spectrometry (LC–MS/MS) analysis, the peptides were separated by a 120 min gradient elution at a flow rate 0.300 µL/min with the Thermo EASY-nLC1200 integrated nano-HPLC system. This system was directly interfaced with the Thermo Q Exactive HF-X mass spectrometer. The analytical column, a homemade fused silica capillary column (75 µm ID, 150 mm length; Upchurch, Oak Harbor, WA, USA), was packed with C-18 resin (300 A, 3 µm, Varian, Lexington, MA, USA). Mobile phase A consisted of 0.1% formic acid, whereas mobile phase B consisted of 80% acetonitrile and 0.1% formic acid. The MS was operated in the data-dependent acquisition mode using the Xcalibur 4.1 software. A single full-scan MS in the Orbitrap (400–1800 m/z, 60,000 resolution) was followed by 20 data-dependent MS/MS scans at 30% normalized collision energy. Each MS was analyzed using the Thermo Xcalibur Qual Browser and Proteome Discovery to search the database.

### 2.10. Ubiquitomics

Cells in culture were treated with MG132, a proteasome inhibitor, 12 h before cell harvesting. The proteins were extracted, reduced, alkylated and digested overnight by trypsin. The ubiquitinated peptides, which show GG motif on Lys residues after trypsin digestion, were enriched by the PTMScan Ubiquitin Remnant Motif Kit (Cell Signaling Technology, #5562). After desalting by stage tip, the enriched peptides were injected into LC–MS/MS using a 120 min gradient separation.

### 2.11. Data Extraction from TCGA and Survival Analysis

The RNA-seq transcriptomic data for the Cancer Genome Atlas (TCGA) project on kidney renal cell carcinoma [11] were downloaded from cBioportal. RNA-seq reads were normalized and log2 transformed. Gene signatures expression scores were computed as the geometric mean signature expression. For survival analyses, the signature score was divided into two groups based on peak values of the density curve. 

### 2.12. Statistical Analysis

All statistical analyses were performed using the R software (v3.4.4). For survival analyses, Cox regression was performed using the R survival package. Data were displayed as mean ± standard deviation (SD). *p* < 0.05 was considered statistically significant. For in vitro assays (such as Western blot, immunoprecipitation, colony formation), experiments were reproduced with consistency at least twice. Representative results are shown in the figures. 

## 3. Results

### 3.1. VHL Restoration in 786-O-Depleted HIF2α and Abrogated Orthotopic Tumor Formation

To identify potential new targets of VHL, we focused on the human ccRCC cell line 786-O, which has no *VHL* expression and is commonly used to study ccRCC [41,42]. We restored VHL expression with a retroviral vector expressing HA-tagged human VHL protein [34] (Figure 1A). Consistent with the role of VHL in negatively regulating the HIF pathway, VHL overexpression diminished HIF2α expression and dampened HIF target genes such as GLUT1 and VEGFA (Figure 1A). HIF1α is not expressed by 786-O. There was no significant difference in the 2D colony or 3D sphere formation abilities between 786-O-Ctrl and 786-O-VHL (Figure 1B), consistent with previous reports showing the moderate impact on 786-O proliferation in vitro by VHL restoration [42]. To test the effect on in vivo tumorigenicity, we stably labeled 786-O-Ctrl and 786-O-VHL with a thymidine kinase-GFP-luciferase fusion protein (triple-reporter, TR) [35] and injected the derived cells into the kidneys of athymic female nude mice. Orthotopic tumor formation was monitored through longitudinal bioluminescence imaging (BLI), which showed a drastic loss of tumorigenicity by VHL overexpression (Figure 1C,D). There were weak persistent BLI signals in the kidneys injected with 786-O-VHL-TR cells, but H&E staining was not sensitive enough to detect those residual tumor cells (Figure 1E). A similar result of VHL-induced loss of tumorigenicity was observed when male nude mice were used as the host (Figure 1F,G). These results confirmed the critical role of VHL in ccRCC and indicated that the derived sublines were valid models for studying VHL.

### 3.2. VHL Restoration Downregulated HIF-Driven Pathways

To find VHL-dependent gene expression regulation at both the RNA and protein levels, we performed RNA-seq and mass spectrometry proteomic profiling of 786-O-Ctrl and 786-O-VHL cells (Figure 2A). From the RNA-seq analysis, 592 genes showed significant differential expression (|fold change| > 4, FDR < 0.01) (Figure 2B, Appendix A). These included the downregulation of some previously known HIF targets in 786-O-VHL, for example, the cytohesin 1 interacting protein (CYTIP), which is important for ccRCC metastasis [43]. Applying Gene Set Enrichment Analysis (GSEA) to the differentially expressed genes, we found that multiple known ccRCC-activated pathways were enriched in 786-O-Ctrl cells, including hypoxia, glycolysis, mTORC1 signaling and E2F targets, whereas the pathways enriched in 786-O-VHL included interferon α response, KRAS signaling and apoptosis (Figure 2C, Appendix A). On the other hand, 6704 proteins were detected by LC–MS/MS in 786-O-Ctrl and 786-O-VHL cells, among which 118 proteins showed differential expression (|fold change| > 1.5, FDR < 0.05) (Figure 2D, Appendix A). GSEA showed that malignancy-associated pathways such as hypoxia, glycolysis, epithelial–mesenchymal transition (EMT) and inflammatory response were enriched in 786-O-Ctrl cells (Figure 2E, Appendix A). Fewer pathways were enriched in 786-O-VHL cells, among which fatty acid metabolism was particularly interesting (Figure 2E) because in the CPTAC study of ccRCC, the downregulation of fatty acid metabolism was among the top pathways associated with the loss of chromosome 3p where *VHL* resides [23]. Mechanistically, HIF drives lipid deposition and cancer progression in ccRCC via attenuation of fatty acid metabolism, and VHL restoration promotes fatty acid metabolism in 786-O cells [44]. 

The acquisition of the transcriptome of 786-O-Ctrl and 786-O-VHL provided us with the possibility to test the hypothesis whether there was a concordance of overall transcriptome between 786-O-VHL and VHL^wild type^ (VHL^WT^) RCC cell lines or tumors, and between 786-O-Ctrl and VHL^mutant^ (VHL^mut^) RCC cell lines or tumors. We plotted ccRCC cell lines available in the Cancer Cell Line Encyclopedia (CCLE) [45] as well as 786-O-Ctrl and 786-O-VHL on the two-dimensional space using the linear discriminant analysis (LDA) of their transcriptome and labeled their VHL status. It was clear that 786-O-Ctrl resembled VHL^mut^ ccRCC cell lines instead of VHL^WT^ ccRCC cell lines, whereas 786-O-VHL exhibited the opposite trend (Figure 2F). GSEA of the differentially expressed genes between VHL^mut^ and VHL^WT^ ccRCC cell lines showed that pathways such as hypoxia, glycolysis and E2F targets were enriched in VHL^mut^ cell lines, whereas KRAS signaling and apoptosis were enriched in VHL^WT^ cell lines (Figure 2G). Consistent with the hypothesis above, this pattern is similar to the transcriptomic enrichment result of 786-O-Ctrl and 786-O-VHL, respectively (Figure 2C). We applied a similar analysis to ccRCC TCGA samples and observed a similar segregation of the tumors into VHL^mut^ and VHL^WT^ groups, and, again, 786-O-Ctrl clustered with VHL^mut^ tumors, whereas 786-O-VHL clustered with VHL^WT^ tumors (Figure 2H). GSEA of the differentially expressed genes between VHL^mut^ and VHL^WT^ ccRCC tumors showed that pathways such as EMT and inflammatory response were enriched in VHL^mut^ tumors, whereas fatty acid metabolism was enriched in VHL^WT^ tumors (Figure 2I). This pattern is similar to the proteomic enrichment result of 786-O-Ctrl and 786-O-VHL, respectively (Figure 2E). These results from CCLE and TCGA samples reinforce that VHL is a key regulator for ccRCC development and restoring VHL expression in 786-O is a valid approach to study the function of VHL in ccRCC. 

When upregulated or downregulated genes at the mRNA level or protein levels were mapped on a four-way Venn diagram (Figure 3A), we noted that 289 genes showed consistent downregulation at the RNA and protein levels upon VHL restoration (Figure 3B, Appendix A). Many of these genes were likely driven by HIF signaling. On the list were some of the known HIF targeting genes, such as *SLC2A1* (encoding GLUT1), *ENO2*, *ALDOA*, *LDHA* and *IGFBP3*. In an unbiased manner, when these genes were analyzed for over-represented conserved transcription factor binding sites in the promoter regions with oPOSSUM [46], HIF1A:ARNT was the top enriched transcription factor (Figure 3C). These genes enriched for pathways regulated by HIF signaling, such as glycolysis, hypoxia, E2F targets and mTORC1 signaling (Figure 3D,E, Appendix A). However, we also noticed that two recently reported HIF-independent VHL targets, ZHX2 [21] and TBK1 [47], were only downregulated by VHL overexpression at the protein level but not at the mRNA level (Figure 3F). Overall, these results confirm the central function of the VHL–HIF pathway to regulate hypoxia signaling, metabolism and other malignancy traits, and also highlight the difficulty of only using transcriptomics and proteomics to identify VHL targets. Evidently, a more direct approach is required to facilitate the identification of VHL targets, which are regulated through VHL-dependent ubiquitination and degradation. 

### 3.3. Ubiquitome Profiling Identified Potential VHL Substrates

As an essential component of the Cul2-Rbx1-EloBC-VHL E3 ubiquitin ligase complex, VHL mediates the recognition and lysine ubiquitination of target proteins. When VHL is restored in 786-O, endogenous VHL targets are expected to be ubiquitinated, which may be detected through ubiquitome profiling. To identify ubiquitinated peptides, we used a label-free qualitative method based on the enrichment using an anti-K-ε-GG antibody to analyze 786-O-Ctrl and 786-O-VHL cells in the presence of the proteasome inhibitor MG132, followed by LC–MS/MS (Figure 4A). This method of Gly-Gly (diGly) remnant affinity purification represents a major breakthrough to increase the sensitivity and coverage of ubiquitination detection [48]. By blocking the proteolytic activity of the 26S proteasome complex, MG132 stabilizes the proteins marked by VHL-mediated ubiquitination, thus, facilitates the detection of ubiquitinated peptides. We detected 3840 and 7042 peptides containing lysine ubiquitination (K^ub^) sites in 786-O-Ctrl and 786-O-VHL cells, respectively (Figure 4B). Notice the almost doubled number of peptides from 786-O-VHL cells, consistent with the function of VHL to promote ubiquitination. In the same experiment, we also profiled cells without MG132 treatment, and the detected peptide numbers dropped abruptly to 456 for 786-O-Ctrl and 626 for 786-O-VHL, indicating that most of the ubiquitinated proteins would be degraded through the 26S proteasome. 

From Figure 4B, the 3873 ubiquitinated peptides unique to 786-O-VHL cells (Appendix A) represented 2026 proteins (Appendix A). The length of these peptides was distributed between 7 and 31 amino acids, in accordance with the property of tryptic peptides with one miss-cleavage (Figure 4C). To gauge if these proteins may contain bona fide VHL substrates, we overlapped the 2026 proteins with the 219 VHL-interacting proteins (Appendix A) curated at the proteins–proteins interaction (PPI) database of the IntACT databases [49]. We saw highly significant overlap of 63 proteins that corresponded to 132 peptides (Figure 4D, Appendix A), including well-characterized VHL targets such as HIF2α (EPAS1), FN1 and VHL itself [50,51,52,53]. This result suggests that our ubiquitome approach should be valid to enrich for putative VHL substrates. 

To identify potential VHL substrates using both proteome and ubiquitome data, we plotted the 2026 ubiquitinated proteins unique to 786-O-VHL cells in a volcano plot with the fold change and FDR values of these proteins from the proteome dataset and found that 426 proteins were both ubiquitinated and significantly downregulated by VHL restoration in 786-O (Figure 4E, Appendix A). Pathway enrichment analysis of these 426 proteins using Metacore software indicated that NF-κB signaling was among the most enriched pathways (Figure 4F). VHL loss is known to activate the NF-κB pathway through HIFα accumulation [32,54,55]. Our results suggest that VHL loss may also activate the NF-κB pathway through losing control of certain NF-κB pathway components as direct VHL substrates (see below).

### 3.4. Potential VHL Substrates with Clinical Prognostic Significance

To prioritize on the most clinically relevant candidates from the 426 potential VHL targets, we overlapped them with the proteins upregulated in ccRCC tumors in comparison to paired normal adjacent tissues (NAT) based on the CPTAC database [23], which resulted in 57 overlapped proteins (Figure 5A, Table 1). Because CPTAC database does not have information about patient outcome, we used the TCGA database of ccRCC [11] and stratified patients into high or low expression groups of these 57 genes. Higher expression of this gene signature was significantly associated with shorter overall survival, shorter progression-free survival and shorter disease-free survival (Figure 5B). Using the transcriptome data (GSE85258) from a study that profiled patient-matched primary and pulmonary metastatic ccRCC tumors [56], we found that this signature was expressed at a higher level in metastases (Figure 5C). Using the chromosome 3p status as a surrogate for *VHL* copy number status, we segregated the CPTAC patients into 3p loss, copy neutral and copy neutral with LOH (loss of heterozygosity) groups and calculated the signature expression at the protein level. For the tumors, 3p loss and copy neutral LOH samples expressed the signature at a significantly higher level than copy neutral samples, but this was not observed for the matched normal samples (Figure 5D). Importantly, the mRNA level of the 57-gene signature displayed no difference between VHL^WT^ and VHL^mut^ tumors in the ccRCC TCGA cohort (Appendix A), suggesting the influence of VHL on this signature is through posttranslational regulation, despite the prognostic significance of this signature at the transcript level. 

Due to the implication of mTOR signaling in ccRCC progression and the use of mTOR inhibitor everolimus for the treatment of advanced ccRCC, we used the FunRich tool [57] to build the protein–protein interaction network between the 57 proteins on our list and the 52 proteins in the KEGG mTOR pathway. We observed that at least 12 out of 57 proteins in our list formed connections with the signaling players in the mTOR pathway (Appendix A). Moreover, we analyzed the transcriptome of patients treated with everolimus in a study that compared nivolumab with everolimus [58] and found that patients with higher expression of the 57-gene signature displayed worse progression-free survival (Appendix A). Therefore, there is a potential that a number of the proteins in the 57-gene list are involved in mTOR signaling in ccRCC and even the response of ccRCC to mTOR inhibitors.

We used two methods to trim down the 57-gene signature to a shorter gene list for better prognostic significance. The random forest method trimmed the list to 21 genes, and ranking genes based on correlation *p* values with overall survival trimmed the list to 20 genes (Appendix A). Both shorter lists outperformed the 57-gene list based on prognosis of overall survival, progression-free survival and disease-free survival of the ccRCC TCGA patients and the 20-gene list showed the highest hazard ratio (Appendix A). 

### 3.5. TGFBI and NFKB2 Are Putative VHL Targeted Proteins

Among the proteins on the list, we performed a number of experiments to validate two candidates, TGFBI and NFKB2. We confirmed the downregulation of TGFBI by VHL restoration in 786-O (Figure 5E). More importantly, we identified K676 in the peptide LAPVYQK^ub^LLER as a new ubiquitination site for TGFBI (Figure 5F). Clinically, the TGFBI protein is upregulated in ccRCC tumors (Figure 5G) and a higher TGFBI transcript level is associated with worse overall survival (Figure 5H). Similarly, we showed that VHL restoration in 786-O led to a lower NFKB2/p100 level (Figure 5I). Two lysine ubiquitination sites were identified, K72 in YGCEGPSHGGLPGASSEK^ub^GR and K741 in GHTPLDLTCSTK^ub^VK (Figure 5J). Because these ubiquitination events were only detected in 786-O-VHL cells, they may be mediated by the Cul2-Rbx1-EloBC-VHL E3 ubiquitin ligase complex. Clinically, the NFKB2 protein is upregulated in ccRCC tumors (Figure 5K) and a higher NFKB2 transcript level is associated with worse overall survival (Figure 5L). To examine if VHL regulates TGFBI and NFKB2 in other ccRCC cell lines, we generated VHL-overexpressed sublines of two other VHL-defective ccRCC cell lines, RCC4 and RCC10, and showed that VHL restoration in these cells also led to the lower expression of TGFBI and NFKB2/p100 at the protein level (Figure 5M).

Next, MG132 treatment abolished the reduction of TGFBI and NFKB2 by ectopic VHL expression in 786-O (Figure 6A), consistent with the hypothesis that VHL directs TGFBI and NFKB2 for proteasome degradation. For TGFBI, we confirmed that 786-O-VHL cells secreted less TGFBI than 786-O-Ctrl cells (Figure 6B). Functionally, 786-O-VHL cells with lower TGFBI expression showed weaker adhesion to fibronectin compared with 786-O-Ctrl cells (Figure 6C). 

More experiments were performed to validate NFKB2 as a putative VHL substrate because it was not reported before. First, we plotted the normalized protein levels of VHL and NFKB2 based on the ccRCC CPTAC database and observed a significant inverse correlation (Figure 6D). Next, we used immunoprecipitation (IP) followed by immunoblot to show that there was an association between NFKB2 and VHL in 786-O cells (Figure 6E). Finally, we used HEK293T (immortalized human embryonic kidney cells) that could be robustly transfected and ectopically overexpressed VHL, NFKB2 (wild type) and/or NFKB2 (K72R mutant) in order to examine if we could recapitulate VHL-mediated degradation of NFKB2 in a manner dependent on at least one of two ubiquitination sites identified in this study (Figure 6F). HEK293T cells expressed low endogenous VHL and a moderate level of endogenous NFKB2 (lane 1). VHL overexpression reduced the endogenous NFKB2 level (lane 1 and 3). Co-transfection of VHL and NFKB2^WT^ increased the NFKB2 level but not as high as transfecting NFKB2^WT^ alone (lane 2 and 4), presumably due to the effect of VHL on destabilizing NFKB2^WT^. Transfection of NFKB2^K72R^, regardless of VHL co-transfection, increased the NFKB2 level to a higher level, similar to transfecting NFKB2^WT^ alone (lane 5, 6 and 2). Importantly, this is the first time that K72 of NFKB2 is shown to affect NFKB2 stability in a VHL-dependent manner. 

## 4. Discussion

In this study, we applied genome-wide transcriptomics, proteomics and large-scale ubiquitomics on 786-O-Ctrl and 786-O-VHL cells to determine the molecular impact of VHL restoration on ccRCC cells and identify potential VHL substrate proteins. Among the omics techniques, ubiquitome was particularly useful, because it is based on the biochemical function of VHL, and it helped to pinpoint the exact ubiquitination sites in VHL substrate candidates. Our study is the first to integrate proteome and ubiquitome of VHL-restored cells to identify VHL substrates. To enhance the clinical relevance of the findings, we also integrated the ccRCC TCGA and CPTAC data during analysis.

Our study has made several significant contributions. First, using orthotopic injection of luciferase-labeled 786-O sublines, we confirmed that persistent extinction of VHL expression is required for ccRCC tumor maintenance (Figure 1). While previous studies proved that VHL restoration in 786-O inhibited tumor formation in subcutaneous models [42,59,60,61], our study confirmed this phenomenon in the orthotopic site. This result has strong clinical implications because restoring the expression of tumor suppressor genes such as p53 has long been considered as a plausible approach for cancer treatment, based on the studies showing tumor regression after p53 restoration in preclinical models [62]. In the case of VHL, gene therapy aimed at regaining VHL expression with delivery methods such as plasmid and adenovirus was shown to retard tumor growth in mice [63,64]. Recently, a new gene therapy method for tumor suppressor restoration, the synthetic mRNA nanoparticle, was invented to restore p53 and PTEN and showed impressive anti-tumor efficacy [65,66]. It will be exciting to apply this technique to restoring VHL as a new form of treatment for ccRCC. We should note that our result showed a low level of persistent tumor signals in the kidney despite VHL restoration (Figure 1E), suggesting that VHL gene therapy, even if it is developed, may still require a combination with other therapeutic modalities for complete tumor eradication. We speculate that the residual tumorigenic potential may reflect the oncogenic activities from mutations in other tumor suppressor genes (e.g., PTEN, TP53, CDKN2A) and oncogenes (e.g., TERT promoter) in 786-O that might cooperate with VHL mutation in initial ccRCC development in patients.

Second, transcriptomic and proteomic profiling of 786-O-Ctrl and 786-O-VHL cells confirms the predominant role of VHL in antagonizing HIF signaling and the tumor-promoting pathways that are centrally regulated by HIF (Figure 2 and Figure 3). Among the pathways enriched in 786-O-Ctrl (HIF2α-high) in comparison to 786-O-VHL (HIF2α-depleted) (Figure 2C,E and Figure 3D), most can be explained by the known master regulatory function of HIF in numerous signaling processes, such as the hypoxia response, glycolysis, mTOR signaling, Myc and E2F targets, EMT, cholesterol homeostasis, inflammatory response involving the NF-κB pathway, G2M checkpoint and DNA repair (both regulated by p53 which is inhibited by HIF2α) [6,7,67,68]. Regarding the pathways enriched in 786-O-VHL cells, including the interferon α response, KRAS signaling, apoptosis and fatty acid metabolism, insights can be gained from studies in ccRCC or other diseases. Fatty acid metabolism is dysregulated in ccRCC cells favoring intracellular lipid accumulation and resulting in the clear cell phenotype, and the mechanism is through HIFα-dependent repression of the rate-limiting enzyme transporting fatty acid into mitochondria [44]. When VHL restoration diminishes HIF2α, it permits fatty acid transportation and promotes its catabolism. The enrichment of interferon α response after VHL restoration is consistent with the role of HIF2α to mediate the resistance to anti-viral type I interferon response in ccRCC cells [69]. The enrichment of KRAS signaling upon VHL restoration is probably related to the paradoxical function of HIF2α to constrain KRAS-AKT signaling in non-small cell lung cancer driven by Kras^G12D^ [70]. For apoptosis, besides the anti-apoptosis function of HIF2α through restricting p53 activity [71], VHL also directly interacts with and inhibits p53 independent of HIF to regulate apoptosis [72].

At the transcriptomic level, in both ccRCC cell lines and TCGA tumors, 786-O-Ctrl clustered with VHL^mut^ cases whereas 786-O-VHL clustered with VHL^WT^ cases (Figure 2F–I), supporting VHL as a key regulator in ccRCC development. Moreover, *Vhl* knockout in *Vhl*-normal murine RCC cell line Renca led to HIFα stabilization, the upregulation of HIF target genes, EMT and enhanced lung metastasis formation [73]. All of these properties are strongly associated with original 786-O cells compared with 786-O-VHL cells in our study. Overall, the majority of the transcriptomic and proteomic changes caused by VHL restoration can be traced back to the diminished HIF2α activity, a point demonstrated by HIF1A:ARNT being the top enriched transcription factor for the genes downregulated at both the transcript and protein levels in 786-O-VHL (Figure 3C). For the HIF-independent VHL targets that may be hidden in the dataset, we argue that integrating the ubiquitome data as an orthogonal approach will be helpful to narrow down the candidates.

Third, through rational integration of ubiquitome (from cell lines) and proteome data (from both cell lines and clinical samples), we generated a list of 57 potential VHL substrates (Figure 5A, Table 1). Various systematic approaches have been developed to identify E3 ubiquitin ligase substrates, such as yeast two-hybrid, in vitro ubiquitination screen, global protein stability profiling, differential expression (shotgun) proteomics, ubiquitin ligase trapping and proximity labeling as well as diGly remnant affinity purification [48]. Previously, a yeast two-hybrid approach with the VHL substrate recognition domain as the bait [74], and a genome-wide in vitro expression approach coupled with a GST-binding screen for VHL substrates [21], identified a number of non-HIFα VHL substrates, such as DGKI, ZHX2 and SFMBT1. In our study, we utilized TMT-labeling shotgun proteomics and diGly remnant-based ubiquitomics, a combined approach that is used for the first time to identify VHL substrates. This method found some known VHL substrates such as HIF2α and FN1. From the substrate candidates, we focused on TGFBI and NFKB2, based on their strong implication in ccRCC progression [25,29] and known regulation by VHL in the case of TGFBI [27,28]. For both proteins, we confirmed lower expression in VHL-restored 786-O cells and showed high-quality MS/MS spectrum of the ubiquitinated peptides (Figure 5). For TGFBI, K676 was ubiquitinated (Figure 5D). Coincidently, TGFBI K676 is acetylated in the blood of COVID-19 patients with severe pneumonia [75]. Because the competition between ubiquitination and acetylation of the same lysine residues is a mechanism to regulate protein stability (e.g., Smad7 [76]), it will be interesting to determine if K676 also regulates TGFBI stability through the competition between these two distinct modifications. For NFKB2, both K72 and K741 were found to be ubiquitinated (Figure 5H). These sites are different from previous a ubiquitination site (K855) modified by the SCF^β-TrCP^ E3 ubiquitin ligase for the maturation from p100 to p52 [31]. We also found NFKB2 K741^ub^ in the ProteomicsDB database with uncharacterized functions [77]. While our results indicate that NFKB2 and VHL physically interact with each other, and K72 of NFKB2 affects NFKB2 stability in a VHL-dependent manner (Figure 6E,F), future studies will more rigorously examine whether these ubiquitinations depend on prolyl hydroxylation and their functional role in ccRCC development.

TGFBI protein upregulation is validated as being associated with metastasis and poorer survival in ccRCC [25,26], but there is no publication yet about the prognostic significance of NFKB2 protein in ccRCC, although other signaling players in the non-canonical NF-κB pathway (NIK, IKKα, and RelB) were shown to correlate with poorer cancer-specific survival [29]. Previous proteomics studies of ccRCC did not have patient survival data [23,78,79,80], so we were unable to generate survival curves for NFKB2 from these studies. We should clearly note that the 57-protein list might contain other VHL target proteins with important functions in ccRCC development. Two promising candidates for future validations based on their known connection to ccRCC are SLC16A3 (monocarboxylate transporter 4), which correlates with poorer relapse-free survival and functionally sustains the Warburg effect and survival in ccRCC cells [81], and RUNX1, which correlates with poorer clinical survival and functionally drives ccRCC [82].

Why are the known VHL-dependent oncoproteins not in the 57-protein list? To answer this question, we would like to highlight that these 57 proteins as VHL substrate candidates have met three criteria (Figure 5A): ubiquitinated in 786-O-VHL cells based on our data; downregulated at the protein level in 786-O-VHL cells based on our data; upregulated at the protein level in ccRCC tumors based on the CPTAC database. First, for the direct substrate targets of VHL, besides HIF1α and HIF2α, there are only a few known non-HIFα VHL substrates, including AKT, fibronectin, collagen IV, AURKA, RPB1, DGKI, ZHX2 and SFMBT1. Many of these proteins have oncogenic roles in ccRCC. However, when we surveyed these proteins in the list of 345 proteins upregulated in the 103 ccRCC samples compared with NAT samples (log2 fold-change > 1; Benjamini–Hochberg adjusted *p* < 0.05) from the CPTAC study [23], none of these known VHL substrates were on the list. This result is somewhat surprising. However, considering the heterogeneity and variations of clinical samples, it is possible that the signals indicative of increased level of these proteins in VHL-defective ccRCC samples were hidden in the noise from the large sample pool. Nevertheless, this fact at least partially explains why none of the known direct targets of VHL showed up in the 57-protein list. Second, for indirect targets of VHL that are upregulated through HIF transcriptional activation and with oncogenic functions, many are gratifyingly among the CPTAC upregulated protein list, for example, Glut1 (SLC2A1), VEGFA, LDHA and HK2. However, these would not make our 57-protein list either because they are not VHL substrates, so they are not on the list of the 2026 ubiquitinated proteins unique to 786-O-VHL cells (Appendix A).

A few limitations are present in our study. First, we applied multiple omics approaches to a single ccRCC model. Even though we have validated the downregulation of TGFBI and NFKB2 by ectopic VHL overexpression in RCC4 and RCC10 (Figure 5M), future studies should validate the VHL candidate substrates across more VHL-defective and VHL-intact ccRCC cell lines. Second, although we argue that our multi-omics approach is a rational method to identify VHL substrates, the false positive rate of the candidate list will require extensive validation to determine. Third, our approach is biased toward identifying ubiquitination substrates fated for proteasomal degradation but unlikely to detect substrates fated for other signaling events. Fourth, because our study only utilized ccRCC cell lines with null *VHL* status, the relevance of the findings in ccRCC with wild type *VHL* is uncharacterized. In the future, the findings from our study, especially the abundance of the 57 proteins as VHL substrate candidates, should be tested through immunohistochemistry in human ccRCC tumors stratified into VHL-defective and VHL-intact subgroups.

Even though the 2019 Nobel Prize in Physiology or Medicine was awarded to the groundbreaking discoveries of VHL and hypoxia response mechanisms, there are still many unanswered questions about VHL, especially the identity of non-HIF targets and their physiological and pathological functions. We believe that our findings and datasets will provide new insight and resources for the mechanistic understanding and therapy discovery of ccRCC and VHL disease.

## Figures and Tables

**Figure 1 cells-11-00472-f001:**
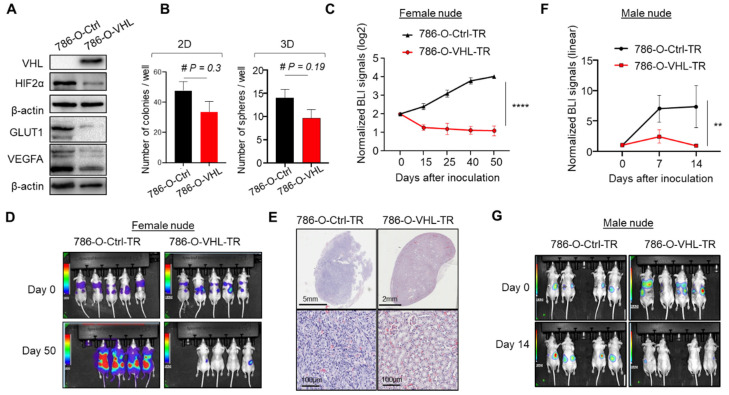
VHL restoration in 786-O-depleted HIF2α and abrogated orthotopic tumor formation. (**A**) Effect of enforced VHL overexpression on HIF2α, GLUT1 and VEGFA levels in 786-O cells, detected by Western blot. (**B**) Comparison of 786-O-Ctrl and 786-O-VHL in forming 2D colonies or 3D tumor spheres. # *p* > 0.05 by Student’s *t*-test. (**C**) Normalized BLI signals for female nude mice orthotopically injected with 786-O-Ctrl-TR (n = 5) or 786-O-VHL-TR (n = 4). BLI signals were normalized to the Day 0 signals and log2 transformed. (**D**) BLI images of Day 0 and Day 50 (endpoint). (**E**) Representative H&E staining of kidneys from mice injected with 786-O-Ctrl-TR or 786-O-VHL-TR. In B and C, data represent mean ± SD. **** *p* < 0.0001 by Student’s *t*-test. (**F**) Normalized BLI signals for male nude mice orthotopically injected with 786-O-Ctrl-TR (n = 7) or 786-O-VHL-TR (n = 6). BLI signals were normalized to the Day 0 signals. Data represent mean ± SD. ** *p* < 0.05 by Student’s *t*-test. (**G**) BLI images of Day 0 and Day 14.

**Figure 2 cells-11-00472-f002:**
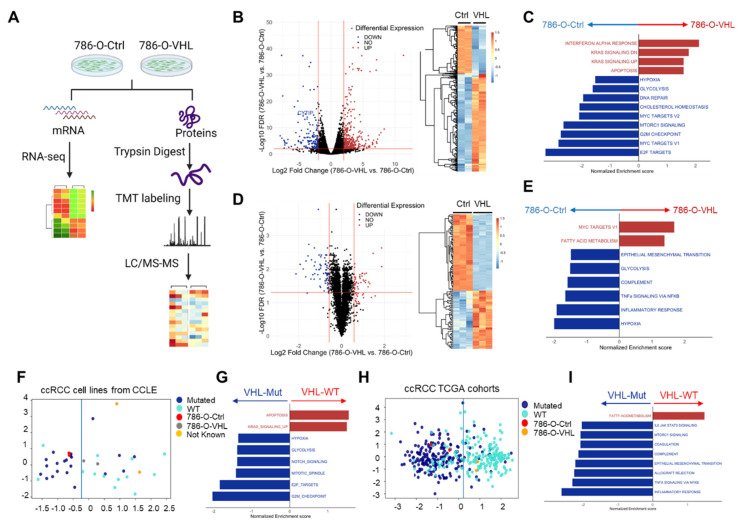
VHL restoration downregulated HIF−driven pathways at both mRNA and protein levels. (**A**) Workflow of RNA-seq and mass spec proteomic profiling in 786-O-Ctrl and 786-O-VHL. (**B**) Volcano plot showing mRNA expression comparison between 786-O-VHL and 786-O-Ctrl (baseline) based on RNA-seq data (left panel). Red lines denote |fold Change| > 4 and FDR < 0.01. Heatmap showing clustering of differentially expressed mRNA (|fold Change| > 4, FDR < 0.01) between 786-O-Ctrl and 786-O-VHL (right panel). (**C**) Top enriched GSEA hallmark pathways (*p* < 0.05, FDR < 0.25) by comparing differentially expressed genes between 786-O-VHL and 786-O-Ctrl. (**D**) Volcano plot showing protein level comparison between 786-O-VHL and 786-O-Ctrl (baseline) based on proteomics data (left panel). Red lines denote |fold change| > 1.5, FDR < 0.05. Heatmap showing clustering of differentially expressed proteins (|fold change| > 1.5, FDR < 0.05) between 786-O-Ctrl and 786-O-VHL (right panel). (**E**) Top enriched GSEA hallmark pathways (*p* < 0.05, FDR < 0.25) by comparing differentially expressed proteins between 786-O-VHL and 786-O-Ctrl. (**F**) Linear discriminant analysis (LDA) plot of CCLE ccRCC cell lines (n = 43) to the two-dimensional transcriptome space with VHL status annotated. 786-O-Ctrl and 786-O-VHL were also drawn on the plot and they fell on the two sides of the boundary line. (**G**) Top enriched GSEA hallmark pathways (*p* < 0.05, FDR < 0.25) by comparing differentially expressed genes between CCLE VHL^mut^ ccRCC cells lines and VHL^WT^ ccRCC cell lines. (**H**) LDA plot of the ccRCC TCGA tumors (n = 403) to the two-dimensional transcriptome space with VHL status annotated. 786-O-Ctrl and 786-O-VHL were also drawn on the plot and they fell on the two sides of the boundary line. (**I**) Top enriched GSEA hallmark pathways (*p* < 0.05, FDR < 0.25) by comparing differentially expressed genes between TCGA VHL^mut^ ccRCC tumors and VHL^WT^ ccRCC tumors.

**Figure 3 cells-11-00472-f003:**
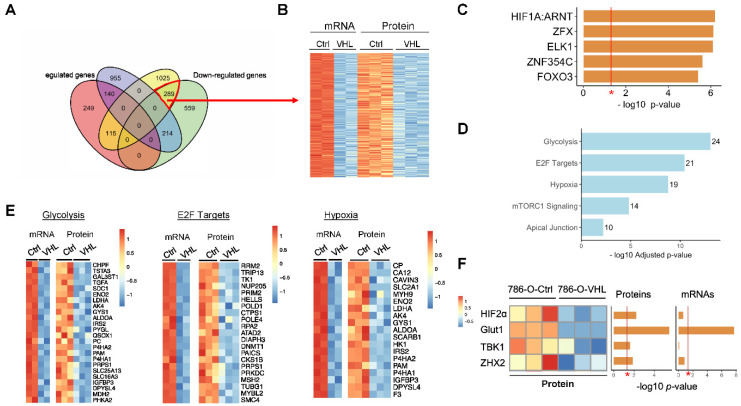
Downregulated genes at both mRNA and protein levels by VHL restoration enriched for HIF binding sites. (**A**) Venn diagram showing the overlap between significantly upregulated or downregulated mRNA and proteins between 786-O-Ctrl and 786-O-VHL (FDR < 0.05). (**B**) Heatmap of significantly downregulated mRNA and proteins between 786-O-Ctrl and 786-O-VHL. Each row represents one gene. (**C**) Top 5 over-represented conserved transcription factor binding sites in the promoter region of downregulated mRNA and proteins analyzed by oPOSSUM. * *p* < 0.05, Fisher’s exact test. (**D**) Top enriched pathways based on Enrichr pathway enrichment analysis for 289 genes downregulated at both mRNA and protein levels by VHL restoration. Numbers denote the differentially represented genes that fall into the pathway. (**E**) Heatmaps showing mRNA and proteins expression between 786-O-Ctrl and 786-O-VHL in glycolysis, E2F targets, and hypoxia pathways. (**F**) Heatmap showing protein expression of HIF2α, GLUT1, TBK1 and ZHX2. * *p* = 0.05.

**Figure 4 cells-11-00472-f004:**
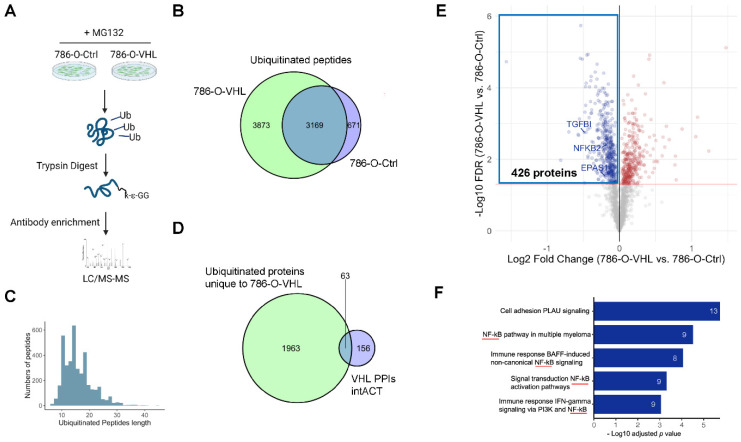
Ubiquitome profiling identified potential VHL substrates. (**A**) Workflow of ubiquitome profiling in 786-O-Ctrl and 786-O-VHL. (**B**) Venn diagram showing the overlap of ubiquitinated peptides between 786-O-Ctrl and 786-O-VHL with MG132 treatment. (**C**) Peptide length distribution of the ubiquitinated peptides unique to 786-O-VHL. (**D**) Venn diagram showing the overlap between ubiquitinated proteins in 786-O-VHL and VHL-interacting proteins in IntACT. (**E**) Volcano plot (FDR < 0.05, red line) of the protein level comparison between 786-O-Ctrl and 786-O-VHL for the 2026 ubiquitinated proteins. Blue box denotes the 426 proteins downregulated and ubiquitinated proteins in 786-O-VHL. (**F**) Analysis of significantly regulated pathways based on the 426 downregulated and ubiquitinated proteins with MetaCore. Numbers denote the proteins that fall in each pathway. Red line underscores NF-κB pathway.

**Figure 5 cells-11-00472-f005:**
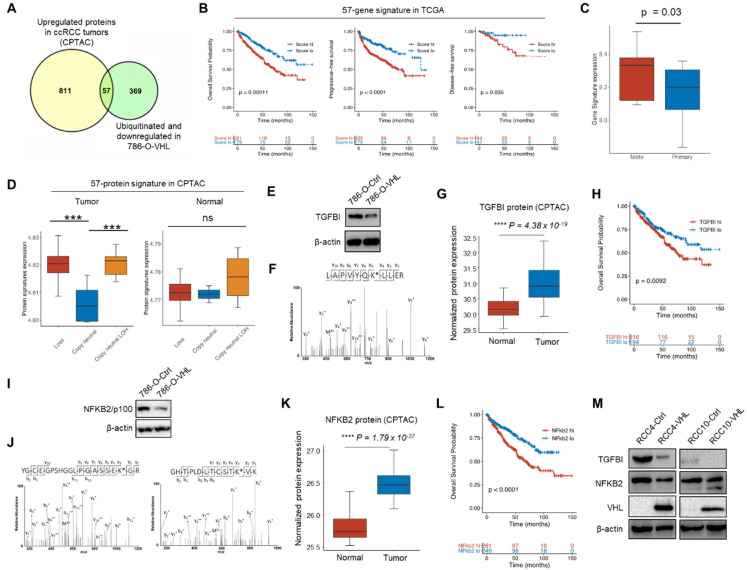
Potential VHL substrates with clinical prognostic significance. (**A**) Venn diagram showing the overlap between the two protein lists. (**B**) Kaplan–Meier analysis for the expression score of the 57-gene signature in relationship with overall survival, progression-free survival and disease-free survival of the TCGA ccRCC patients. *p* values calculated based on Cox regression analysis. (**C**) mRNA expression score of the 57-gene signature in matched primary ccRCC tumors and pulmonary metastases (GSE85258). *p* value calculated by *t* test. (**D**) Protein expression score of the 57-protein signature in the ccRCC CPTAC patient tumors and NAT tissues, stratified based on chromosome 3p status. *** *p* < 0.01,**** *p* < 0.001, ns (not significant, *p* > 0.05), calculated by *t* test. (**E**) Western blot of TGFBI in 786-O-Ctrl and 786-O-VHL cells. (**F**) The MS/MS result of the TGFBI peptide LAPVYQK^ub^LLER containing the ubiquitinated lysine K676, the Δm between y_5_ and y_4_ corresponds to the mass of Lys residue plus diGly. (**G**) TGFBI protein expression in normal tissues and tumors in ccRCC patients based on the CPTAC data. (**H**) Kaplan–Meier analysis for overall survival for patients of the TCGA ccRCC cohorts based on TGFBI mRNA expression. *p* value calculated from Cox regression analysis. (**I**) Western blot of NFKB2/p100 in 786-O-Ctrl and 786-O-VHL cells. (**J**) The MS/MS result of the NFKB2 peptide YGCEGPSHGGLPGASSEK^ub^G containing the ubiquitinated lysine K72 and GHTPLDLTCSTK^ub^VK containing K741. (**K**) NFKB2 protein expression in normal tissues and tumors in ccRCC patients based on the CPTAC data. (**L**) Kaplan–Meier analysis for overall survival for patients of the TCGA ccRCC cohorts based on NFKB2 mRNA expression. *p* value calculated from Cox regression analysis. (**M**) Western blot of TGFBI and NFKB2/p100 in RCC4-Ctrl, RCC4-VHL, RCC10-Ctrl and RCC10-VHL cells.

**Figure 6 cells-11-00472-f006:**
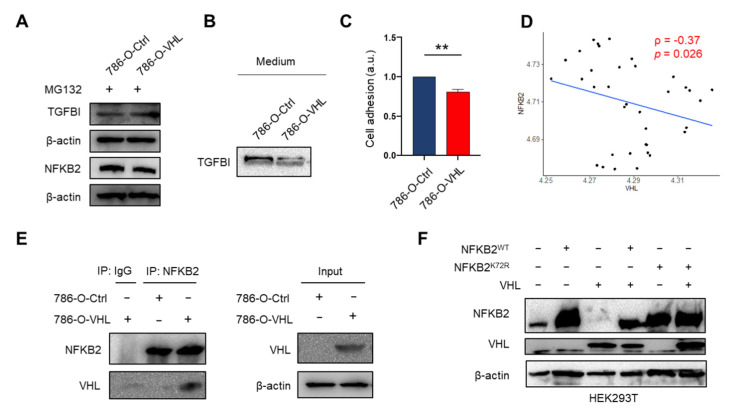
Validation of TGFBI and NFKB2 as putative VHL targeted proteins. (**A**) Western blot of TGFBI and NFKB2 in 786-O-Ctrl and 786-O-VHL cells after MG132 treatment. (**B**) Western blot of TGFBI in the conditioned medium of 786-O-Ctrl and 786-O-VHL cells. (**C**) Cell adhesion assay of 786-O-Ctrl and 786-O-VHL cells to fibronectin-coated plate. ** *p* < 0.01 by Student’s *t*-test. (**D**) Scatter plot of normalized VHL and NFKB2 protein expression with data from the ccRCC CPTAC study. Pearson correlation coefficient ρ and *p* value are shown. (**E**) Immunoassay of the endogenous VHL–NFKB2 association in 786-O-Ctrl and 786-O-VHL cells, assessed by immunoprecipitation (IP) with immunoglobulin G (IgG), as a control, or with anti-NFKB2, followed by immunoblot analysis with anti-NFKB2 or anti-VHL. (**F**) Western blot of NFKB2 and VHL in HEK293T cells transiently transfected with NFKB2 (wild type), NFKB2 (K72R mutant) and/or VHL plasmids.

**Table 1 cells-11-00472-t001:** Overlapped protein list in Figure 5A. These 57 proteins are downregulated and ubiquitinated in 786-O-VHL cells and upregulated in ccRCC tumors compared with paired normal adjacent tissue samples in the CPTAC study. The proteins are ranked by fold change (tumor/normal) in the CPTAC study.

Protein Name	log2 Fold Change (Tumor/Normal)	Adjusted *p* Values (Tumor vs. Normal)	Protein Name	log2 Fold Change (Tumor/Normal)	Adjusted *p* Values (Tumor vs. Normal)
FTL	2.53	4.2 × 10^−12^	HMOX1	0.74	1.2 × 10^−29^
SLC16A3	2.32	1.9 × 10^−52^	RHBDF2	0.74	1.6 × 10^−38^
PLOD2	2.29	7.4 × 10^−33^	PARP9	0.74	3.3 × 10^−38^
PYGL	2.04	2.1 × 10^−43^	SMC4	0.72	2.7 × 10^−45^
SCARB1	1.95	9.8 × 10^−34^	DDX60	0.71	4 × 10^−40^
TGM2	1.62	1.4 × 10^−37^	HM13	0.7	1.4 × 10^−23^
GYS1	1.52	1.7 × 10^−44^	DENND3	0.7	2.1 × 10^−47^
HLA-B	1.26	1.1 × 10^−30^	RNF213	0.7	5.1 × 10^−35^
NEK6	1.24	1.3 × 10^−51^	NFKB2	0.67	1.1 × 10^−39^
DPP9	1.12	6 × 10^−53^	APAF1	0.67	9.5 × 10^−53^
APOL2	1.11	2.8 × 10^−24^	RRP1	0.64	3.1 × 10^−53^
ERGIC1	1.09	1.1 × 10^−35^	SRM	0.64	6.9 × 10^−33^
UBE2L6	1.07	1 × 10^−61^	CAD	0.63	5.5 × 10^−56^
OAS3	1.02	5.3 × 10^−44^	SLC39A14	0.61	2 × 10^−10^
ALDOA	1.01	7.6 × 10^−57^	HELZ2	0.61	1.4 × 10^−27^
PLEKHA2	0.98	4.1 × 10^−49^	TBC1D2	0.61	7.8 × 10^−29^
MYO9B	0.97	9.6 × 10^−69^	CNDP2	0.6	3.9 × 10^−23^
IMPDH1	0.95	8.3 × 10^−49^	CDK17	0.59	7.3 × 10^−45^
TGFBI	0.93	2.4 × 10^−15^	GFPT1	0.59	1.9 × 10^−29^
TRIM22	0.92	8.9 × 10^−49^	ARHGEF1	0.59	4.5 × 10^−55^
EHD2	0.88	8 × 10^−28^	NAP1L1	0.57	3.7 × 10^−45^
HLA-C	0.87	1.7 × 10^−19^	IPO9	0.56	4.1 × 10^−64^
RUNX1	0.87	2.7 × 10^−37^	MTHFD2	0.54	4.8 × 10^−16^
ANXA2	0.86	7.7 × 10^−49^	ASCC3	0.54	3.9 × 10^−57^
PARP14	0.85	3.8 × 10^−48^	COL6A2	0.52	5.1 × 10^−16^
FNDC3B	0.81	3.8 × 10^−46^	TBC1D2B	0.51	2.5 × 10^−36^
ASNS	0.78	1.7 × 10^−27^	PPIP5K2	0.51	1.1 × 10^−41^
AMPD2	0.76	1.3 × 10^−38^	GSDMD	0.51	1.3 × 10^−41^
LAMB2	0.75	1.1 × 10^−26^			

## Data Availability

The RNA-seq data were deposited to GEO under the accession number GSE186013. All other data that support the findings of this study are available from the corresponding authors upon reasonable request.

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
