# Peer review of "Multi-Omics Profiling to Assess Signaling Changes upon VHL Restoration and Identify Putative VHL Substrates in Clear Cell Renal Cell Carcinoma Cell Lines"

_cells, 2022, doi:10.3390/cells11030472_

Round 1

Reviewer 1 Report

This manuscript describes that RNA-seq, mass spec proteomic profiling, and ubiquitome profiling identified reduction of TGFBI and NFKB2 in the VHL-restored 786-O model. This manuscript shows preliminary results.

This study requires functional assays of TGFBI and NFKB2 as VHL-dependent ubiquitination substrates.

Two lysine ubiquitination sites (line 377) should be experimentally verified.

Reviewer 2 Report

Re: review

The purpose of this study was to evaluate the effect of VHL restoration at the molecular level in a single cell line model. While this approach may be reasonable and potentially beneficial for cancer research, this report suffers from a number of weaknesses, as described below. 

Major comments:

As the authors acknowledge, their investigation was performed on a single cell line which make very difficult to interpret the overall conclusions. First of all, the tile of the manuscript should be renamed as “Multi-omics Profiling to Assess Signaling Changes upon VHL 2 Restoration and Identify Putative VHL Substrates in Clear Cell in a Human Renal Cell Carcinoma Cell Line”.

The main conclusion of the study “restoring VHL expression almost completely abolishes orthotopic tumor formation in immuno-deficient mice” should be discussed carefully. One of the main hurdle of the in vivo modelling in RCC (mouse) is the paucity of VHL-mutated cell lines. Majority of the cell lines studied in the literature (RAG, Renca in syngeneic models; Caki-2, ACHN in immune compromised mice) are VHL normal and all mice uptake the tumors with very aggressive tumor behavior. Thus, what is actually the role of VHL in tumor development?

Moreover, if their hypothesis is correct, that the reversal hypothesis should work. Practically, any RCC cell line with normal (“restored”) VHL should behave at the level of multi-omics as the restored VHL 786-O cell line. Upon, KO the VHL that cell line should behave as WT 786O with defective VHL. That will be a very critical experiment to validate their hypothesis.

786-O is not a deficient HVL cell line, is a mutated VHL cell line. This is not discussed. Moreover, 786-O is also known as cell lines with mutations in CDKN2A, PTEN TP53 (2 mutations) and TERT, all genes so important in tumorigenesis. Also, they have interactions with VHL. This is also not discussed.

There are many speculative associations between the observed data.  For example restoring VHL was correlated with upregulation of fatty acid metabolism gene. Is that a fact generated by experimental observation or a consequence of stats. It is very well known that in clear cell RCC that excess of fatty acid metabolism leads to HIF activation which is VHL independent.

Minor comments:

Results section contain extensively information which are supposed to be presented in the Discussion section.

Many information missed the corresponding reference.

There is no information related to the reproducibility of the experiments.

As a small note also, the spheroids assay should have be performed U not flat bottom plates.

In vivo experiments should have be performed in male also.

Taken together, all the above potential biases, significantly limit the interpretation of the findings and challenge the validity of the conclusions made.  Thus, a resubmitted manuscript is needed. This manuscript cannot be accepted without major revisions and thorough new experiments.

Reviewer 3 Report

This is a nicely presented multi-omics approach for assessing the role of VHL restoration in ccRCC. The authors need to be congratulated on their comprehensive efforts. However, as a major comment, it would be advisable to repeat some of their major experiments in a different RCC cell line in order to suggest that they are more generalizable.

Reviewer 4 Report

To gain deeper understanding of VHL function and find other VHL targets  for ccRCC and overcoming the resistance to HIF2α inhibitor the authors have profiled and integrated transcriptome, proteome and ubiquitome of VHL-deficient ccRCC, 786-O cell line, after restoring VHL expression. The data support that the predominant effect of VHL restoration is the downregulation of HIF-related signaling and metabolic pathways. By co-analysis of data and the Clinical Proteomic Tumor Analysis Consortium (CPTAC) study of ccRCC, 57 proteins were identified that are ubiquitinatedand downregulated by VHL restoration. Higher expression of this gene signature was significantly associated with worse overall survivalin ccRCC patients. Among the 57 proteins, VHL restoration significantly reduced the protein expression of TGFBI and NFKB2, both of which are likely important players in promoting ccRCC.

The study is well performed, but too focused on descriptive data. If not incorporated in the study, at least, needs an in deep discussion with the relevant signalling in oncogenicity in these tumors. Many VHL-dependent proteins with strong oncogenic driver activity have been related to ccRCC, and they are not in the signature. It must be discussed, and what is the possibility that other proteins in the signature are drivers instead these others. Without this, it looks as other work with no real relevance in ccRCC clinic handeling.

As used as biomarkers is fine, but this needs to be pursued in deep in the study. Is really the signature true marker of worse prognosis or residual disease, or metastasis, or DFS? Must be shown. Is a derived, shorter signature, better predictor? mTOR inhibitors are being used in this tumors, is the signature related to some extent in this response? How does the signature relate to the 10% of VHL positive tumors?

The fact that NFKb2 or TGFBI are predictors can be related to other tuimorigenic signalling not related to VHL, especially if mRNA levels are plotted. Fig 5 F and J, KMs must be of protein, not mRNA.

Comparison to real tumors VHL+ and VHL- in ccRCC should be discussed. There are several proteomic analysis of ccRCC tumors directly from patients that should be discussed accordingly in this work.

Identifications of A few limitations present in the study is a good point, but as it is is limited. Most important, not considered, is that this is done in a cell line (artificial culture, the proper cell line, not the most representative of ccRCC, besides being the most used, mutational drift in the cell line...). Comparison to real tumors VHL+ and VHL- in ccRCC should be discussed (see above)  and informed as limitation.

Round 2

Reviewer 2 Report

I thank the authors for their great work to address to all the questions/comments. The updated manuscript is responsive to my previous comments. I think that this updated manuscript is appropriate for publication now.

Author Response

We appreciate the reviewer for agreeing that our responses were accepted and the manuscript now is appropriate for publication. 

Reviewer 3 Report

No additional comments to be addressed.

Author Response

We appreciate the reviewer for agreeing that our responses were accepted and raising no further questions. 

Reviewer 4 Report

The authors reply well to all comments. The manuscript is improved.

Author Response

We appreciate the reviewer for agreeing that our responses were accepted and commending on the improved quality of the manuscript.